# Nonspecific Amyloid Aggregation of Chicken Smooth-Muscle Titin: In Vitro Investigations

**DOI:** 10.3390/ijms24021056

**Published:** 2023-01-05

**Authors:** Alexander G. Bobylev, Elmira I. Yakupova, Liya G. Bobyleva, Nikolay V. Molochkov, Alexander A. Timchenko, Maria A. Timchenko, Hiroshi Kihara, Alexey D. Nikulin, Azat G. Gabdulkhakov, Tatiana N. Melnik, Nikita V. Penkov, Michail Y. Lobanov, Alexey S. Kazakov, Miklós Kellermayer, Zsolt Mártonfalvi, Oxana V. Galzitskaya, Ivan M. Vikhlyantsev

**Affiliations:** 1Institute of Theoretical and Experimental Biophysics, Russian Academy of Sciences, Pushchino, 142290 Moscow Region, Russia; 2A.N. Belozersky Institute of Physico-Chemical Biology, Lomonosov Moscow State University, 119991 Moscow Region, Russia; 3Institute of Protein Research, Russian Academy of Sciences, Pushchino, 142290 Moscow Region, Russia; 4Institute for Biological Instrumentation, Federal Research Center, Pushchino Scientific Center for Biological Research of the Russian Academy of Sciences, Pushchino, 142290 Moscow Region, Russia; 5Department of Early Childhood Education, Himeji-Hinomoto College, 890 Koro, Kodera-cho, Himeji 679-2151, Japan; 6Institute of Cell Biophysics, FRC PSCBR, Russian Academy of Sciences, Pushchino, 142290 Moscow Region, Russia; 7Department of Biophysics and Radiation Biology, Semmelweis University, 1085 Budapest, Hungary; 8Institute of Fundamental Medicine and Biology, Kazan Federal University, 420008 Kazan, Russia

**Keywords:** smooth muscle titin, protein aggregates, amyloid aggregation, amyloids, cross-β

## Abstract

A giant multidomain protein of striated and smooth vertebrate muscles, titin, consists of tandems of immunoglobulin (Ig)- and fibronectin type III (FnIII)-like domains representing β-sandwiches, as well as of disordered segments. Chicken smooth muscles express several titin isoforms of ~500–1500 kDa. Using various structural-analysis methods, we investigated in vitro nonspecific amyloid aggregation of the high-molecular-weight isoform of chicken smooth-muscle titin (SMT_HMW_, ~1500 kDa). As confirmed by X-ray diffraction analysis, under near-physiological conditions, the protein formed amorphous amyloid aggregates with a quaternary cross-β structure within a relatively short time (~60 min). As shown by circular dichroism and Fourier-transform infrared spectroscopy, the quaternary cross-β structure—unlike other amyloidogenic proteins—formed without changes in the SMT_HMW_ secondary structure. SMT_HMW_ aggregates partially disaggregated upon increasing the ionic strength above the physiological level. Based on the data obtained, it is not the complete protein but its particular domains/segments that are likely involved in the formation of intermolecular interactions during SMT_HMW_ amyloid aggregation. The discovered properties of titin position this protein as an object of interest for studying amyloid aggregation in vitro and expanding our views of the fundamentals of amyloidogenesis.

## 1. Introduction

It is known that, to normally perform their biological functions, newly synthesized proteins must fold into a certain three-dimensional structure [1,2]. The folded structures of proteins are only moderately stable. Certain factors, such as genetic mutations or disturbances of protein synthesis and degradation, can lead to incorrect protein stacking or misfolding followed by the formation of pathological aggregates [3]. Improper protein stacking is a rather common phenomenon associated most often with the development of diseases such as amyloidoses [4]. With the development of amyloidosis in humans or animals, the protein loses its native conformation to form amyloids, aggregates of incorrectly folded forms of protein having a specific structure. Amyloid aggregates, as well as their intermediate forms (mainly oligomers), may lead to cellular death [5,6,7,8,9,10].

Amyloid aggregates of many proteins have been found in various tissues and organs of amyloidosis humans and animals. These diseases include, in particular, liver amyloidosis, Alzheimer’s disease, Parkinson’s disease, type II diabetes, and prion diseases, as well as systemic amyloidoses [11,12,13,14]. Amyloids have a number of specific characteristics, such as the ability to bind to Congo Red and thioflavin T; they are resistant to proteases and insoluble in most solvents [14]. The main property of amyloids, regardless of the type of aggregates they form, is the presence of a quaternary cross-β structure [15,16].

Besides pathological amyloids, there are functional amyloids—protein aggregates that perform certain functions. Functional amyloids are found in many species of living organisms from plants, fungi, and protozoa to higher animals, including humans [17,18,19,20,21,22,23,24,25,26,27].

Despite the extensive investigation, there is as yet no clear understanding of the molecular mechanisms of amyloid aggregation. The differences behind the mechanisms of the formation of functional versus pathological amyloids have yet to be resolved. This is probably due to the complexity of the object of study. In order to obtain the required information about amyloids, it is necessary to use an array of methods (such as nuclear magnetic resonance spectroscopy, transmission electron microscopy (TEM), cryo-electron microscopy, atomic-force microscopy (AFM), X-ray diffraction, small-angle X-ray scattering (SAXS), Fourier-transform infrared spectroscopy (FTIR), which is often technically difficult to perform, since each of the methods has its own limitations regarding the object of study. 

Titin (also called connectin [28,29]) is one of the most difficult objects when studying its aggregation properties. In vitro studies on isolated titin preparations revealed a tendency of this protein to form various sorts of intermolecular interactions that led to the formation of oligomers and aggregates. Thus, the aggregation of titin in vitro was first investigated in 1993 [30]. It was shown that, in solutions of low ionic strength (0.1 M KCl near neutral pH (6.5)), striated muscle titin assembles into higher-order aggregates [30]. In 2003, atomic-force microscopy revealed that, in a solution containing 25 mM imidazole–HCl, 0.2 M KCl, 4 mM MgCl_2_, 1 mM EGTA, 0.01% NaN_3_, 1 mM dithiothreitol (DTT), 20 µg/mL leupeptin, 10 µM E-64, pH 7.4, isolated titin molecules are capable of self-assembly into oligomers within a mere 10 min [31]. The observed titin oligomers could be divided into bimolecular species, as well as consisting of several molecules. In each case, however, a globular head was observed, with which other titin molecules interacted, thus forming an oligomer [31].

In vitro investigations of the specificity of aggregation between titin domains conducted in 2005 concluded that the ability for aggregation increases when the identity of amino acid sequences of the domains is greater than 40% [32]. In 2015, the in vitro aggregation of some titin Ig-domains obtained by a recombinant method was studied [33]. The neighboring identical titin domains were shown to be able to form misfolded structures [33].

The experiments in vitro with molecular simulations carried out in 2015 showed that during the refolding of tandem repeats (I27–I27 and I27–I28 immunoglobulin-like domains of titin), independent of sequence identity, more than half of all molecules transiently formed a wide range of misfolded conformations [33]. Simulations suggested that a large fraction of these misfolds resemble an intramolecular amyloid-like state named “intramolecular amyloids” [33]. These authors also reported that, during the refolding of tandem repeats, the development of a strand-swapped long-lived misfolded state without an amyloid-like structure is possible [33].

In 2016, the ability of a low-molecular-weight (500 kDa) isoform or possibly a truncated form of chicken smooth-muscle titin (SMT_LMW_) to form in vitro amyloid aggregates was described [34]. In 2018, the ability of that isoform to form in vitro amyloid aggregates with different properties in two solutions differing in ionic strength was found [35].

In the present work, using a number of structural analysis methods, we investigated the formation of an in vitro amyloid structure of a chicken smooth-muscle titin isoform with a molecular weight of ~1500–1600 kDa (SMT_HMW_). A peculiar feature of this protein and, in particular, this isoform of titin relative to other amyloid proteins studied is its large size. One titin molecule is comparable to the size of the protofibrils of some amyloid proteins. Thus, it was of interest to elucidate whether this high-molecular-weight isoform of smooth-muscle titin was capable of forming aggregates, their structural features, and the type of aggregation—amyloid/non-amyloid. The results obtained point at the nonspecific amyloid aggregation of the protein.

## 2. Results

### 2.1. DLS Analysis of SMT Aggregations 

Based on our previous research into SMT_LMW_ [34,35] and SMT_HMW_ [36], we were aware of the conditions required to form amyloid aggregates of these proteins. This work, in order to understand the features of chicken titin aggregation rate, investigated the kinetics of SMT_HMW_ aggregation. We used dynamic light scattering (DLS), the simplest and most convenient method to study the aggregation kinetics of amyloid proteins [34,35,37]. The method also enables analyzing the aggregation reversibility, which has been shown earlier for SMT_LMW_ [35].

Figure 1a shows a change in the autocorrelation function of the scattered light upon the formation of SMT_HMW_ aggregates in a solution containing 0.15 M glycine–KOH at pH 7.0–7.5 over 60 min. During the first 20 min of incubation, the correlation function *g*_1_(*t*) was observed to decay almost mono-exponentially (Figure 1a), with the subsequent emergence of a shoulder at high correlation times. This is indicative of the formation of large aggregates with smaller diffusion coefficients. After 40 min, the correlation function *g*_1_(*t*) featured a greater decay time and a more pronounced shoulder (Figure 1a) at high correlation times, which indicated an aggregation increase. A pronounced correlation-function shoulder was also observed during 40, 50, and 60 min of incubation (Figure 1a).

Several peaks reflecting the dimensions of SMT_HMW_ aggregates were obtained by the correlation function analysis. Prior to the formation of aggregates, only two well-resolved peaks with the average *R*_h_ of approximately 16 nm (the dominating peak of approximately 73%) and 86 nm (the minor peak of approximately 27%) were observed (Figure 1b, 0 min). The first peak corresponds, most likely, to SMT_HMW_ molecules, whereas the second can be indicative of their insignificant aggregation. During the first 20 min of incubation, we observed a shift of the peaks and an increase in the ratio of the volumes of the larger- and smaller-size fraction, indicating the development of the titin aggregation process (*R*_h_~53 nm and ~479 nm) (Figure 1b). 

After 40 min of the experiment, a stable fraction of larger aggregates with a hydrodynamic radius of ~1175 nm (96.7%) appeared. This fraction was also observed after 50 and 60 min of the experiment (Figure 1b). Another peak, indicating the formation of titin aggregates with the average *R*_h_ of approximately 3813 nm emerged after 50 min of incubation and, after over 60 min, *R*_h_ was more than 4475 nm. It should be noted that the peak with *R*_h_ of approximately 4500 nm was at the limit of the range for this method. Therefore, the formation of larger SMT_HMW_ aggregates cannot be excluded (Figure 1b).

The next task was to elucidate the ability of SMT_HMW_ aggregates to disaggregate upon increasing ionic strength. Figure 1c presents data on the partial disaggregation of SMT_HMW_ aggregates after incubation in a solution containing 0.6 M KCl and 30 mM KH_2_PO_4_ (pH 7.0). A decrease in the percentage of particles with larger hydrodynamic radii and an increase in the number of particles with smaller hydrodynamic radii (Figure 1c) were observed. 

### 2.2. Electron and Atomic-Force Microscopy of SMT_HMW_ Aggregates

Based on the DLS data presented above, investigations were conducted using two time intervals (1 h and 24 h). Figure 2 shows AFM images of SMT_HMW_ molecules in a solution of 0.6 M KCl, 30 mM KH_2_PO_4_, 1 mM DTT, 0.1 M NaN_3_, and pH 7.0. It was found that, as in the case of striated-muscle titin [31,38,39], the smooth-muscle titin molecules had a filamentous shape with threads 3–4 nm thick and ~300 nm (on average) long with a globular head on one end (Figure 2a, inside the white squares). Besides individual molecules, oligomers of this protein were also found: several titin molecules interacting with one another, often forming a thickening in the center (Figure 2b). Similar oligomers have been found earlier in titin preparations isolated from rabbit skeletal muscles [31].

Figure 3 shows electron micrographs of negatively stained SMT_HMW_ aggregates formed in a solution containing 0.15 M glycine–KOH, pH 7.0–7.5, at 4 °C. SMT_HMW_ formed amorphous aggregates after 1 h and 24 h incubation (Figure 3). Aggregates became larger after 24 h of incubation. In a number of cases, filamentous structures of about 4 nm in diameter (represented, most likely, by titin filaments) were observed between amorphous aggregates (Figure 3c). 

According to AFM, after 1 h, aggregation SMT_HMW_ aggregates looked like large amorphous structures several micrometers long and up to 350 nm high (Figure 4a). After 24 h of aggregation, they had the form of branching flattened structures of more than 10 μm in length and 200–250 nm in height (Figure 4b,c). 

To determine their capability of disaggregation, SMT_HMW_ aggregates formed during 1 and 24 h were transferred to conditions with increased ionic strength (0.6 M KCl, 30 mM KH_2_PO_4_, 1 mM DTT, 0.1 M NaN_3_, and pH 7.0). Figure 4d presents the results of 1 h disaggregation. As can be seen in the figure, SMT_HMW_ became much smaller and spherical amorphous aggregates of about 100–200 nm in diameter and 100–120 nm in height. Apart from amorphous aggregates, individual filaments occurred on the substrate (Figure 4g); judging by their size, they may be bundles of SMT_HMW_ molecules.

Figure 4e,f shows the disaggregation results for SMT_HMW_ aggregates formed for 24 h. Spherical amorphous aggregates up to 23 nm in height can be seen, as well as a network of threads with a height of 5–6 nm. 

### 2.3. Circular Dichroism 

Figure 5a illustrates the circular dichroism (CD) spectrum of SMT_HMW_ before and after the formation of aggregates. No changes in the secondary structure were detected upon the formation of SMT_HMW_ aggregates: after chromatography the preparation had 6 ± 6% α-helices and 41 ± 6% β-structures, 53 ± 6% of turns and a disordered structure, whereas aggregated SMT_HMW_ had a helix and β-structure content of 5 ± 6% and 42 ± 6%, respectively, 53 ± 6% of turns and disordered structure. Thus, in both cases, we detected a high content of β-structure and a disordered secondary structure. 

### 2.4. Fourier-Transform Infrared Spectroscopy 

Figure 5b presents FTIR data obtained at 20 °C and corrected for the spectral contribution of water vapor and CO_2_. The experimental data were analyzed following the principles described in [40]. The obtained estimates of the content of secondary structure elements in samples of titin and its aggregates are given in Table 1. As the circular dichroism data, the FTIR data indicate that the secondary structure of the protein does not change during SMT_HMW_ aggregation. 

### 2.5. Association of SMT_HMW_ Aggregates with Thioflavin T

To identify the amyloid nature of SMT_HMW_ aggregates, we investigated their binding to thioflavin T. A significant increase in ThT fluorescence intensity in the presence of SMT_HMW_ aggregates formed over 1 h (Figure 5c) and 24 h (Figure 5d) was recorded and compared with that in the presence of monodispersed SMT_HMW_. After disaggregation, the ThT fluorescence in the presence of SMT_HMW_ aggregates was at the same level as that of monodispersed SMT_HMW_ after 1 and 24 h of the experiment.

### 2.6. X-ray Diffraction of SMT_HMW_ Aggregates

The amyloid cross-β structure of SMT_HMW_ aggregates was revealed by X-ray diffraction (Figure 5e,f). A 1 h aggregation featured a diffuse reflex at ~10 Å and a relatively sharp reflex at ~4.7 Å (Figure 5e). X-ray diffraction of SMT_HMW_ aggregates after a 24 h aggregation revealed a diffuse reflex at ~10 Å and a sharp reflex at ~4.8 Å (Figure 5f). The detected reflections can be ascribed to a cross-β structure. Thus, the presence of a cross-β structure identified by X-ray diffraction analysis confirms that SMT_HMW_ aggregates are amyloids.

### 2.7. Small-Angle X-ray Scattering (SAXS)

SAXS was used to obtain information about the conformation of molecules in solution. For such high-molecular-weight structures as titin, this method makes it possible to approximately estimate their internal conformation, taking into account the tangent of the slope (tan *A*) of the log *I*–log *S* dependence. It is known that, for a rod-shaped conformation, tan *A* = 1; that, for a planar conformation, tan *A* = 2; that, for globular particles tan, *A* = 4. Using the DAMMIF program [41], it is also possible to visualize the approximate three-dimensional structure of the protein.

From these data (Figure 6), it follows that molecular SMT_HMW_ has a flat shape (plate conformation) (tan *A* = −2.7) (Figure 6a). The latter is clearly visible in the insert (Figure 6c). The aggregated shape of SMT_HMW_ particles is also close to plate-like (more charged plate conformation) (tan *A* = 2.65) (Figure 6b).

### 2.8. Differential Scanning Calorimetry of SMT_HMW_

The thermal stability of SMT_HMW_ aggregates was elucidated by differential scanning calorimetry (DSC). Figure 7 presents typical temperature dependences of excess heat capacity of SMT_HMW_ in monomeric and aggregated forms. As seen in the figure, the curves have heat absorption peaks that might correspond to cooperative disruption of the structure. Repeated heating of the preparations confirms that the process is irreversible. The heat absorption peak maximum temperature of molecular titin, *T*_m_, was 317.7 ± 0.1 K; the value of transition calorimetric enthalpy, Δ*H*_cal_ = 870 ± 90 kJ mol^−1^. The heat absorption peak maximum temperature of titin aggregates, *T*_m_, was 321.7 ± 0.1 K; the value of transition calorimetric enthalpy, Δ*H*_cal_ = 1090 ± 110 kJ mol^−1^. 

### 2.9. SMT Amino Acid Sequence Identity

To reveal segments with large amino acid sequence identity that, according to available data, have an increased tendency for aggregation [33], we calculated the identity in the amino acid sequence between adjacent pairs of domains in smooth-muscle titin. Chicken titin (UniProtKB—A6BM71_CHICK) was chosen for the calculations. Calculations carried out using the BLAST program showed that the average identity in the amino acid sequence between neighboring FnIII domains does not exceed 33 ± 7%, and between neighboring Ig domains, it does not exceed 20 ± 11%, which is a relatively low indicator (Table 2, Appendix A). In two cases, however, domains with an identity higher than 50% were observed.

### 2.10. Calculation of Unstructured Areas in the SMT Molecule

The disorder was revealed by analyzing the amino acid sequence UniProtKB—A6BM71_CHICK using IsUnstruct, a specialized program for predicting the natural disorder of proteins [43,44]. The average disorder of a titin fragment was 90% for PRK12323; 90%, Ig-like; 30%, FnIII-like; 55%, none (Figure 8). 

## 3. Discussion

In the present work, we investigated the amyloidogenic propensity in the smooth-muscle isoform of the giant protein titin. According to our results, three titin isoforms—with molecular weights of ~500, ~1200, and ~1500–1600 kDa—were isolated from chicken gizzard muscle tissue. Previously, we have shown the 500 kDa splice-isoform of titin or its truncated fragment (SMT_LMW_) to form in vitro aggregates with amyloid properties and structure [34,35]. In this study, we show that the smooth-muscle titin isoform with MW~1500–1600 kDa and an isoform or a proteolytic fragment of this protein with MW~1200 kDa (SMT_HMW_) also form aggregates in vitro. Detailed research into the SMT_HMW_ aggregation process was conducted to better understand the changes occurring in this protein, which is undoubtedly involved in smooth muscle contraction. 

Using DLS, we found that, upon decreasing ionic strength, chicken SMT_HMW_ formed large aggregates with a hydrodynamic radius of ~4500 nm over 1 h (Figure 1a). EM and AFM showed SMT_HMW_ aggregates formed over 1 h to be amorphous (Figure 3 and Figure 4). In general, chicken SMT_HMW_ aggregation was fast, which made it impossible to determine the lag period. The high aggregation rate of SMT_HMW_ (three times as high as that for SMT_LMW_ [34]) was, apparently, due to the oligomeric forms of the protein and its monomers present, as shown by AFM, in the high ionic-strength solution (Figure 2). Our results are consistent with the literature data showing that the presence of oligomeric forms at the initial stage of aggregation accelerates the process [45].

An increase of ThT fluorescence detected at the binding of the dye to SMT_HMW_ aggregates an hour after their formation indicates the amyloid nature of the structures formed (Figure 5c,d). This is also supported by the X-ray diffraction data: the presence of reflexes at 4.7 and 10 Å confirms the amyloid nature of SMT_HMW_ aggregates formed both after 1 and 24 h of incubation (Figure 5e,f). The X-ray diffraction data are indicative of the presence of a quaternary cross-β structure in SMT_HMW_ aggregates. It should be noted that both reflexes are circular and partially blurred. Nevertheless, the reflex representing the distance between the beta segments in the sheet was observed both after 1 h (4.7 Å, Figure 5e) and after 24 h (4.8 Å, Figure 5f) of aggregation. The 10 Å reflex representing the distance between the beta sheets was blurred but was also present in both cases. According to numerous literature data, such reflexes are characteristic of amyloid or amyloid-like structures [46]. 

An uncharacteristic feature of the amyloid aggregation of titin is the formation of aggregates without changing the secondary structure, which was confirmed by CD and FTIR independent techniques (Figure 5a,b, Table 1). According to the literature data*,* in vitro experiments with some proteins have shown that prior to the formation of amyloids, the structure of their molecules must undergo a transformation of the type of α-helix to β-folding or random coil to β-sheet [47,48,49]. We revealed no such changes in the amyloid aggregation of chicken SMT_HMW_. Similar results have been obtained earlier for chicken SMT_LMW_ [34,35] and for skeletal myosin binding protein C [46] that also consists of Ig-like and FnIII-like domains. It appears that the ability to form amyloid aggregates without changes in the secondary structure of molecules is a characteristic feature of the above-mentioned multidomain proteins, which distinguishes them from most other amyloid proteins. 

Taking into account the EM data on protein filaments between amorphous SMT_HMW_ aggregates (Figure 3c) and the data on the presence of protein “filaments” after disaggregation of SMT_HMW_ aggregates (Figure 4e–g), we suggested that not the entire protein but only its particular domains were involved in intermolecular interactions in the process of amyloid SMT_HMW_ aggregation. The proposed SMT_HMW_ scheme of stacking to form an amyloid structure during the aggregation is given in Figure 9. In particular, segments that have a disordered structure and those whose structure is amyloid are shown. 

Are there any confirmations of the assumption we make? In the literature, there are comparative data obtained by cryoelectron microscopy on the structure of functional amyloid aggregates of Orb2 and pathological aggregates of amyloid β-peptide [50]. Those authors have shown that only a small part of its molecule is involved in the formation of the amyloid nucleus in the functional amyloids of Orb2, whereas most of the molecule remains dynamically disordered. Formation of the amyloid nucleus in pathological amyloid β-peptide, on the contrary, involves most of its molecule. Given these data and the revealed properties of SMT_HMW_ aggregates, they can be classified as functional, which is indirectly confirmed by their ability to disaggregate with ionic strength increasing (Figure 1 and Figure 4). It should only be noted that this type of aggregation is a model, since neither functional nor pathological SMT_HMW_ aggregates have been found in living cells. 

It should also be understood that, due to the huge size and complex structure of the titin molecule, its complete transition to the amyloid form is hardly possible. However, its particular segments are, most likely, capable of forming an amyloid structure. This assumption can be supported by literature data on the unfolding of individual titin domains [51]. In particular, those authors have shown that the stepwise unfolding/folding of titin immunoglobulin (Ig) domains occurs in the elastic I band region of intact myofibrils at physiological sarcomere lengths and forces of 6–8 pN [52]. For this reason, we consider the formation of amyloid segments to occur exactly between partially opened domains of neighboring titin molecules. 

It has also been shown that most proteins have amyloidogenic segments [53,54]. We calculated the number of amyloidogenic segments in randomly selected domains of a chicken titin fragment, predicted using the FoldAmyloid program (three Ig-like and three FnIII-like). From the data obtained (Appendix A), there are at least two amyloidogenic segments in each of the counted domains; furthermore, for example, the Ig-like (I10) domain has five such segments. These data indicate a high potential propensity of titin, inherent in its domains, to form amyloid aggregates. 

When discussing the type of SMT_HMW_ aggregates, it is necessary to dwell in more detail on the DSC data, which are indicative of yet another feature of titin’s amyloid aggregation. It is known that amyloid fibrils or aggregates of several proteins melt and dissociate at temperatures of the order of 75–100 °C, which manifests itself in the form of characteristic endothermic transitions on the thermograms [55,56,57,58,59]. It has been shown that amyloid fibrils are more resistant to temperature than the native protein (their transition temperatures differ by about 30–40 K) [59]. At the same time, it has been shown that the enthalpy of the melting of amyloid fibrils is less than that of the native protein. Based on this, it has been concluded that the density of intermolecular interactions in the amyloid structure is lower than in the native protein [59]. Our experiments showed that the melting point of the aggregated form of SMT_HMW_ is slightly higher (by only 4 K) than that of the molecular form of the protein (Figure 7). Herewith, the melting enthalpies of the aggregated and non-aggregated forms of SMT_HMW_ practically did not differ. These data suggest that the total binding energy during the formation of SMT_HMW_ aggregates did not change. Attention should be also paid to the heat absorption peaks themselves. If we compare the data obtained by us for SMT with the data for globular proteins, then the heat absorption peak should be much larger. There is a possibility that the current peak of heat absorption can correspond to the energy spent on the breakdown of the aggregates or oligomers of the molecular and aggregated forms of the protein or even its particular domains. If this is the case, then we failed to register the melting point, and it is above 373 K. Nevertheless, the obtained data are of interest. It can be summed up that, in this temperature range (up to 373 K), the molecular form of the protein and its aggregates have relatively similar stability. 

Discussing the issue of the type of (functional or pathological) amyloid aggregation, it is worth noting that no cases of the transformation of functional to pathological amyloids have been recorded. Most probably, there are molecular mechanisms of protection against such a transformation that developed in the process of evolution. Thus, it is known that, in multidomain proteins with homologous tandem repeats, the neighboring domains have a low identity in the amino acid sequence. This feature formed, apparently, as a result of evolutionary pressure, prevents incorrect protein stacking and subsequent aggregation [33]. It is generally accepted that a low tendency for aggregation is characteristic of those proteins in which the identity between domains is less than 40% [33]. Data on the identity of individual titin domains in the amino acid sequence are known [33]. We calculated the identity of amino acid sequences for chicken titin (Table 2). Calculations showed this fragment to contain only two domains with an identity greater than 50% (Appendix A). The average identity of the amino acid sequence between pairs of neighboring FnIII-like or Ig-like domains does not exceed 40% (Table 2). Thus, it can be concluded that chicken SMT has a low tendency towards aggregation. It has been shown, however, that upon sarcomere elongation, titin domains are capable of unfolding in situ [52], herewith opening the hidden hydrophobic sites that may lead to its aggregation. 

In conclusion, regarding the possible role of the revealed changes for the muscle in vivo, it is necessary to recall one of the early works in which the authors show, using AFM, that repeated mechanical cycles of the extension/relaxation of multidomain proteins can lead to the formation of misfolded structures formed by two neighboring domains [60]. The misfolding was completely reversible and changed the mechanical topology of the domains, while maintaining the same stability as in the original folded state. The authors conclude that multidomain proteins can assume a new state of incorrect stacking. These data and the data we obtained can be important from the viewpoint of a better understanding of the functioning of multidomain muscle proteins, in particular titin, in the sarcomere and muscle as a whole. It is not to be ruled out that structural changes occurring in this protein during the muscle extension/contraction cycle are involved not only in the fine-tuning of elastic properties but also in changing the contractile response of the muscle. Future in vivo studies of titin structural modifications and misfolding will reveal more subtle nuances of the involvement of this protein in the functioning of muscle cells.

## 4. Conclusions

In summary, in vitro nonspecific amyloid aggregation of chicken SMT_HMW_ was found. Over relatively short times (within 1 h), the protein formed amorphous amyloid aggregates with a quaternary cross-β structure without undergoing changes in the secondary structure. The thermal stability of SMT_HMW_ aggregates did not practically differ from that observed in protein preparations containing both monomers and oligomers. Amyloid aggregates of SMT_HMW_ disaggregated almost completely at an increase in the ionic strength of the solution. Amyloid aggregates of this type are functional rather than pathological, so a similar aggregation of titin in vivo to perform certain functions cannot be ruled out. However, the involvement of this protein in the formation of amyloid deposits has not been shown yet, which positions this protein as a model object to study the process of the amyloid aggregation of a non-pathological type. Our data expand the views about the fundamentals of amyloidogenesis. Besides, disclosing the mechanisms of the formation of functional and pathological amyloids and understanding their differences at the structural level can give an idea of how the cell regulates amyloid aggregation for its desired functioning, avoiding the manifestation of the toxicity of pathological amyloids. 

## 5. Materials and Methods 

### 5.1. Purification of Chicken Gizzard SMT 

Smooth-muscle titin was prepared from chicken gizzard by the method described in [61] with our modifications described in [34]. SMT_HMW_ was purified by gel filtration on a Sepharose-CL2B column equilibrated in a buffer containing 0.6 M KCl, 30 mM KH_2_PO_4_, 1 mM DTT, 0.1 M NaN_3_, pH 7.0. Protein concentration was determined by a SPECORD UV VIS spectrophotometer using the extinction coefficient (*E*_280_^1 mg/mL^) of 1.37 for titin [62]. For this research, the protein has been isolated more than 20 times. 

### 5.2. SDS-PAGE and Mass Spectrometry Analysis of Titin 

The presence of SMT_HMW_ in the sample was confirmed by sodium dodecyl sulfate–polyacrylamide gel electrophoresis (SDS-PAGE) (Appendix A) and mass spectrometry analysis, the data of which are described in [36]. The molecular weight of SMT_HMW_ was assessed by TotalLab software v1.11 (Appendix A). Two protein bands are visible, which are most likely SMT_HMW_ isoforms, or the lower band is a proteolytic fragment of this protein (Appendix A, gels 1–4). By the densitometry data, the molecular weight of the upper band of the protein is ~1635 ± 245 kDa; its content is ~68.5%. The molecular weight of the lower band is ~1245 ± 189 kDa; its content is ~31.5%. The SDS-PAGE of titin was performed using a separating gel containing 6.5–7% polyacrylamide prepared as described [62]. The gels were stained with Coomassie Brilliant Blue G-250 and R-250 mixed at a 1:1 ratio. For shotgun mass spectrometry analysis, the sample was solubilized into a buffer (4% sodium dodecyl sulfate in 0.1 M Tris-HCl pH 7.6, 0.1 M dithiothreitol) and incubated for 5 min at 95 °C as described [36,63]. The samples were sonicated (4 × 30 s at 20 W; ME220, Covaris, Woburn, MA, USA), centrifuged (5 min, 16,000× *g*), and the supernatant was collected. The YM-30 filter (Millipore, Ireland) was used for alkylation and trypsinolysis (14 h, 2 μg of trypsin (Trypsin Gold, Promega, Madison, WI, USA)) according to the FASP method [64]. Peptides were desalted using C18 microcolumns and subjected to HPLC–MS/MS analysis using the HPLC Ultimate 3000 RSLCnano system (Thermo Scientific, Waltham, MA, USA) as described [36,63]. 

### 5.3. Conditions for the Formation of SMT Aggregates 

Purified SMT_HMW_ in a column buffer (0.6 M KCl, 30 mM KH_2_PO_4_, 1 mM DTT, 0.1 M NaN_3_, pH 7.0) was used to form aggregates. SMT aggregates (concentration, 0.2–0.4 mg/mL) were formed by dialysis in Sigma-Aldrich cellulose membrane tubing (size, 25 × 16 mm) for 1 and 24 h at 4 °C against a solution containing 0.15 M glycine–KOH, pH 7.0–7.5. In disaggregation experiments, SMT aggregates were dialyzed during 1 and 24 h against a column buffer. 

### 5.4. Dynamic Light Scattering Experiments

DLS experiments were conducted according to a protocol described in [34,35]. For the DLS analysis of SMT aggregation, a protein sample in a buffer containing 0.6 M KCl, 30 mM KH_2_PO_4_, 1 mM DTT, 0.1 M NaNO_3_, pH 7.0, at an initial concentration of 1 mg/mL, was transferred into a solution of 0.15 M glycine–KOH, pH 7.0–7.5, by gradual dilution to a final concentration of 0.1 mg/mL to decrease ionic strength. Further steps were as in [35]. The collected autocorrelation functions were converted into particle-size distributions, using the general-purpose algorithm provided with the ZS Zetasizer Nano (Malvern Instruments Ltd., Malvern, UK) used in this experiment. Particle-size distributions obtained from alternative inversion algorithms yielded comparable results. The dynamic viscosity of the protein solutions determined using an SV-10 Sine-wave Vibro Viscometer (A&D Company Ltd., Tokyo, Japan) at 25 °C was 0.92 cP. This value was taken into consideration when measuring the particle dimensions in SMT samples collected 60 min after the dialysis. The analyzed volume of scattering with a beam cross-section of ~100 µm, accounting for the protein concentration used, contained about 10 billion SMT protein molecules, the signal from which was measured. The correlation function signal accumulated over 15 cycles of 15 s each (the way it is described in [34,35]). The results were obtained from three independent experiments.

### 5.5. Transmission Electron Microscopy

A drop of aggregated protein suspension at a concentration of 0.1 mg/mL was applied to a carbon-coated collodion film (2% collodion solution in amyl acetate (Sigma-Aldrich, St. Louis, MO, USA)) on a copper grid (Sigma-Aldrich, St. Louis, MO, USA) and negatively stained with 2% aqueous uranyl acetate (SPI-Chem., West Chester, PA, USA). Samples were examined under a JEM-100B electron microscope (JEOL Ltd., Tokyo, Japan). Samples obtained from five independent protein isolations were analyzed; many different fields of view were analyzed.

### 5.6. Atomic-Force Microscopy

For AFM measurements, titin aggregates were attached to freshly cleaved mica. An aliquot of titin (10–20 µL) was pipetted onto the mica surface and incubated at room temperature for 10 min. Unbound protein was washed away by extensive rinsing with distilled water, then by blowing gently with a stream of high-purity N_2_ gas. The noncontact mode (alternating current or AC mode) AFM images of titin aggregates bound to the mica surface were acquired with a Cypher ES AFM instrument (Asylum Research, Santa Barbara, CA, USA). Scanning was performed at high set-point values (0.8–1.2 V) to avoid the binding of the sample to the cantilever tip. Silicon nitride cantilevers (Olympus) were used for scanning in air (AC160TS, resonance frequency~300 kHz). At a typical scanning frequency of 0.7–1.4 Hz, we collected 512 × 512 pixel or 1024 × 1024 pixel height-, amplitude-, and phase-contrast images. 

To prepare samples of SMT_HMW_ aggregates for AFM, 2 µL of the protein was transferred to freshly cleaved mica and incubated for 5 min. The sample was then washed three times in a drop of distilled water deionized by a type I Milli-Q system for 30 s and dried in the air. AFM imaging was performed using an AFM Ntegra-Vita microscope (NT-MDT, Russia) in noncontact (tapping) mode in air. The typical scan rate was 0.5–1 Hz. Measurements were carried out using NSG03 cantilevers with a resonance frequency of 47–150 kHz and ensured a 10 nm tip curvature radius. The processing and presentations of AFM images were performed using Nova software 1.0.26 (NT-MDT, Moscow Region, Russia) and Gwyddion 2.4450 software (http://gwyddion.net/download-old.php accessed on 17 April 2018. Experiments were replicated by independently executing the process of protein preparation, their incubation at 37 °C, and sample analysis three times. AFM images of a buffer solution (0.15 M glycine–KOH, pH 7.0–7.5) containing no titin are presented in Appendix A. Samples obtained from three independent protein isolations were analyzed; many different fields of view were analyzed.

### 5.7. Circular Dichroism

SMT_HMW_ was dialyzed for 24 h against a buffer containing 0.15 M glycine–KOH, pH 7.0–7.5. The CD spectra prior to and after SMT_HMW_ aggregation were recorded in a Jasco J-815 spectrometer (JASCO Inc., Tokyo, Japan) using 0.1 cm optical path-quartz cells and wavelengths of 250–190 nm. 

Three repeats of the spectrum were taken for each investigated sample. Data processing and graphical representation were performed in the SigmaPlot program. Based on the absorption spectrum, the exact protein concentration was calculated using the formula: C = ABS_280_/*l/K*_e_ (where ABS_280_ is the absorption value at a wavelength of 280 nm, *l* is the optical path length in cm, *K*_e_ is the extinction coefficient).

Three spectra in the far UV region obtained for the investigated sample were averaged and smoothed in the spectropolarimeter software (Spectra Manager Version 2, Spectra Analysis Version 2.02.06 (Build 1) Spectra Analysis Jasco). A similar procedure was done for the three spectra obtained for the buffer solution. The averaged spectrum of the buffer solution was subtracted from the obtained averaged spectrum of the investigated sample. The value of molar ellipticity [Θ] was calculated by the formula:
[Θ]_λ_ = Θ _λ_ · RMW/*l*·*c*,
where Θ _λ_ is the measured value of ellipticity at the wavelength λ, millidegrees; RMW, the average molecular weight of the residue, calculated from the amino acid sequence; *l*, the optical path length, mm; *c*, protein concentration, mg/mL. 

The secondary structure was calculated using the CONTIN/LL module of the CDPro program [65]. The mean root square deviation (RMSD) according to the CD_PRO_ program did not exceed 6%.

### 5.8. Fourier-Transform Infrared Spectroscopy 

Measurements were carried out on a Thermo Scientific Nicolet 6700 FT-IR spectrometer, equipped with the Smart Proteus accessory with a Peltier cuvette holder, in transmission mode in a cuvette of crystalline calcium fluoride with an optical pathlength of 4 μm, using a liquid nitrogen-cooled MCT detector. Scanning in the wavenumber range from 650 to 4000 cm^−1^; resolution, 1 cm^−1^; averaging over 256 spectra. The device was calibrated according to the manufacturer’s instructions.

The IR spectra of titin preparations’ solutions in a corresponding buffer and the spectra of the buffer itself were measured at 20 °C. The concentration of the protein was 10 mg/mL. The optical path length of the CaF_2_ cuvette was calculated for each measurement based on the optical density of the test sample at 3404 cm^−1^, using the water absorption value at an optical pathlength of 1 µm equal to 0.533 AU, adjusted for the protein concentration in the sample [66]. The optical pathlength of the cuvette was 4.52 ± 0.04 μm. The IR spectrum of the protein preparation was measured twice; the buffer spectrum was also registered twice. The IR spectrum of the buffer (0.6 M KCl, 30 mM KH_2_PO_4_, 1 mM DTT, 0.1 M NaNO_3_, pH 7.0, for molecular SMT; and 0.15 M glycine–KOH, pH 7.0–7.5, for the aggregated form of the protein) was subtracted from each protein spectrum, taking into account the difference in the values of the optical path length in the measurements. Each difference spectrum was adjusted for the spectral contribution of water vapor and CO_2_, followed by the analysis in the wavenumber range of 1725 to 1481 cm^–1^ for the content of secondary structure elements in the protein, following the principles described in [40]. A sample obtained from three different isolations of the protein was used. The obtained estimates of secondary structure elements in the protein were averaged by the results of two measurements. The standard deviations of the values of the secondary structure elements in the protein are given. 

### 5.9. Fluorescence Analysis with Thioflavin T

The amyloid nature of the SMT_HMW_ aggregates was estimated by the intensity of thioflavin T (ThT) fluorescence (1 ThT:5 SMT (*w*/*w*)). Fluorescence was measured at λ_ex_ = 440 nm and λ_em_ = 488 nm using a Cary Eclipse spectrophotometer (Varian, Palo Alto, CA, USA). Four independent series of measurements were carried out. The amyloid nature was assessed from the difference of fluorescence intensity between the non-aggregated and aggregated forms of the protein. This method was used as a control of the amyloid nature of titin aggregates after each isolation of the protein. 

### 5.10. X-ray Diffraction

SMT_HMW_ aggregates for X-ray diffraction analysis were prepared after a 1 h and 24 h incubation at 4 °C in an experimental solution. A sample obtained from three different isolations of the protein was used. Then the aggregates were concentrated up to more than 10 mg/mL by centrifugation at 12,000 rpm for 60 min. Droplets of this preparation were placed between the ends of wax-coated glass capillaries (approximately 1 mm in diameter) separated by a gap of approximately 1.5 mm. Fiber diffraction images were collected using a Microstar X-ray generator with HELIOX optics equipped with a Platinum135 CCD detector (X8 Proteum system, Bruker AXS) at the Institute of Protein Research, Russian Academy of Sciences, Pushchino. Cu Kα radiation, λ = 1.54 Å (1 Å = 0.1 nm), was used. The samples were positioned at a right angle to the X-ray beam using a four-axis kappa goniometer. Different exposures and different oscillation angles were used; the sample itself was irradiated in different orientations.

### 5.11. Small Angle X-ray Scattering 

SAXS experiments were carried out on a small-angle camera of the Photon Factory (Tsukuba, Japan). The protein solution in a thermostatted cuvette with mica windows was irradiated by X-rays of a wavelength 1.503 Å at 23 °C. The distance between the sample and the detector was 2.35 m. The range of the measured scattering vectors is *Q* = 0.008–0.2 Å^−1^ (*Q* = 4π sin θ/λ, where λ is the X-ray radiation wavelength and 2θ is the scattering angle). X-ray scattering was detected by a PILATUS 100K two-dimensional X-ray detector. The shape of the particles was estimated from the tangent of the angle of inclination log *I* on log *Q*, where *I* is the scattering intensity, and *Q* is the scattering vector modulus [67]. Log *I*–log *Q* dependences in SAXS data were approximated by linear regression. Correlation coefficients (*R*^2^) ranged from 0.92 to 0.99. 

### 5.12. Differential Scanning Calorimetry 

DSC measurements were made on a SCAL-1 precision scanning microcalorimeter (Scal Co. Ltd., Pushchino, Russia) with 0.33 mL glass cells at a scanning rate of 1 K/min and under a pressure of 2.5 atm [68]. The experiments were performed in 0.15 M glycine–KOH at pH 7.0–7.5. The protein concentrations were 1.2 mg/mL. The experimental calorimetric traces were adjusted for the calorimetric baseline, and the molar partial heat capacity functions were calculated in a standard manner. The excess heat capacity was evaluated by the subtraction of the linearly extrapolated initial and final heat capacity functions with correction for the difference of these functions by using a sigmoidal baseline [69]. DSC experiments were carried out 2 times for both the monomeric and aggregated SMT_HMW_ forms. The obtained curves coincided in temperature *T*_m_ to an accuracy of 0.1 K. The relative error in determining calorimetric enthalpy, ΔHcal did not exceed 10%. 

### 5.13. Calculation of the Identity of the Amino Acid Sequence and Disordered Regions in the SMT_HMW_ Molecule

The SMT_HMW_ amino-acid sequence identity was calculated by the BLAST program. The data were retrieved from the UniProtKB databases: UniProtKB—A6BM71_CHICK. 

## Figures and Tables

**Figure 1 ijms-24-01056-f001:**
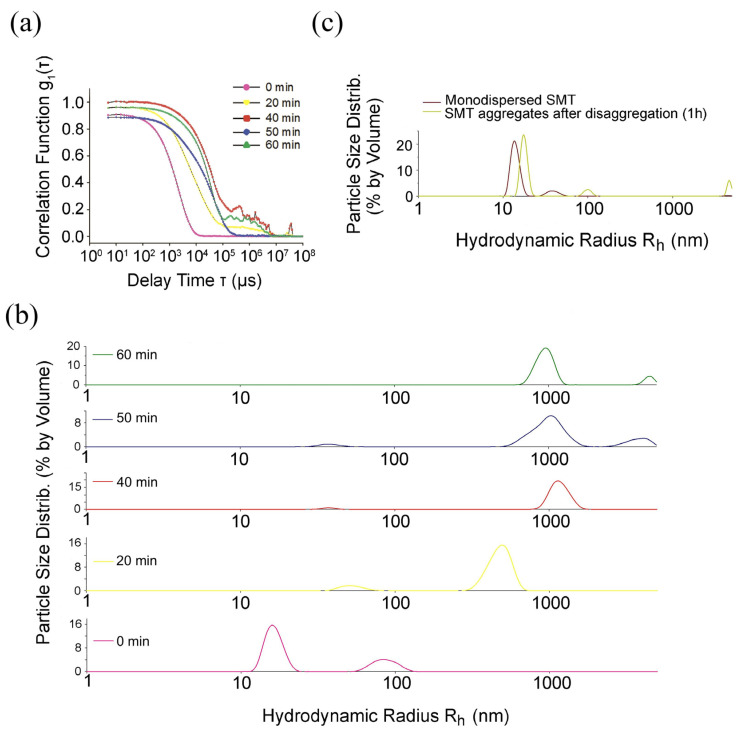
(**a**) Evolution of field autocorrelation functions *g*_1_(*t*) of light scattered during SMT_HMW_ amyloid aggregate formation (pH 7.0–7.5, *T* = 25 °C). Aggregation was observed in a solution containing 0.15 M glycine–KOH, pH 7.0–7.5; (**b**) size distributions of SMT_HMW_ particles. Generation of large aggregates and their time-dependent growth are shown; (**c**) distribution of SMT_HMW_ particles after 1 h disaggregation (for clarity, the graph shows monodispersed SMT_HMW_). The data reflect the distributions corresponding to certain time points. Three independent experiments were carried out.

**Figure 2 ijms-24-01056-f002:**
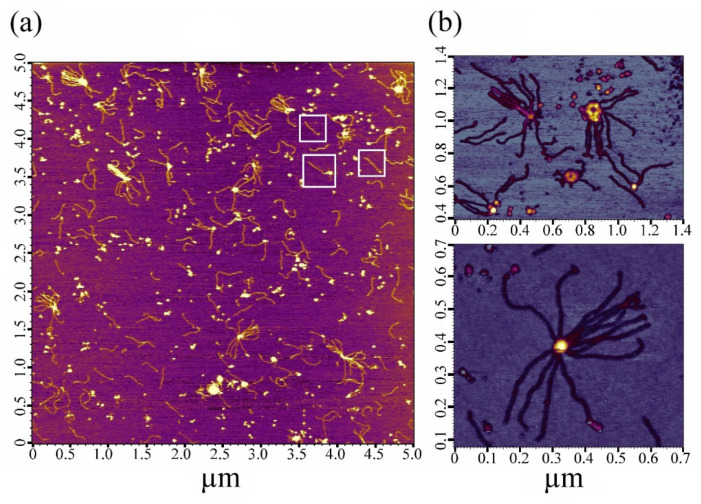
AFM images of molecular SMT_HMW_ obtained in a solution of 0.6 M KCl, 30 mM KH_2_PO_4_, 1 mM DTT, 0.1 M NaN_3_, and pH 7.0. In the field of view, there are filamentous protein molecules—monomers (marked with white squares) (**a**), which assemble into oligomers forming a central thickening (**b**).

**Figure 3 ijms-24-01056-f003:**
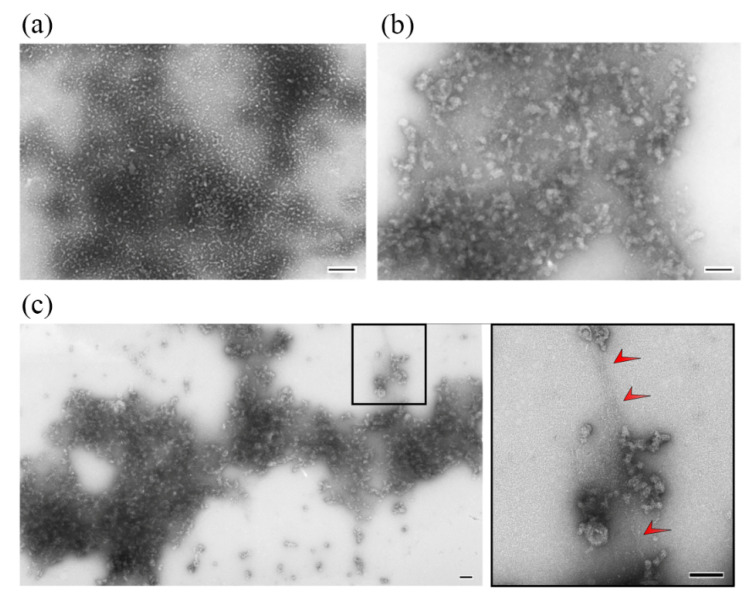
Electron microscopy of SMT_HMW_ aggregates formed in a solution containing 0.15 M glycine–KOH at pH 7.0–7.5 at 4 °C. (**a**) 1 h aggregation of SMT_HMW_. (**b**,**c**) 24 h aggregation of SMT_HMW_. Protein filaments of a diameter of about 4 nm between amorphous aggregates (shown by red arrows on the insert) can be seen. The most representative data, obtained as a result of 10 independent experiments, are given. Scale bar, 100 nm.

**Figure 4 ijms-24-01056-f004:**
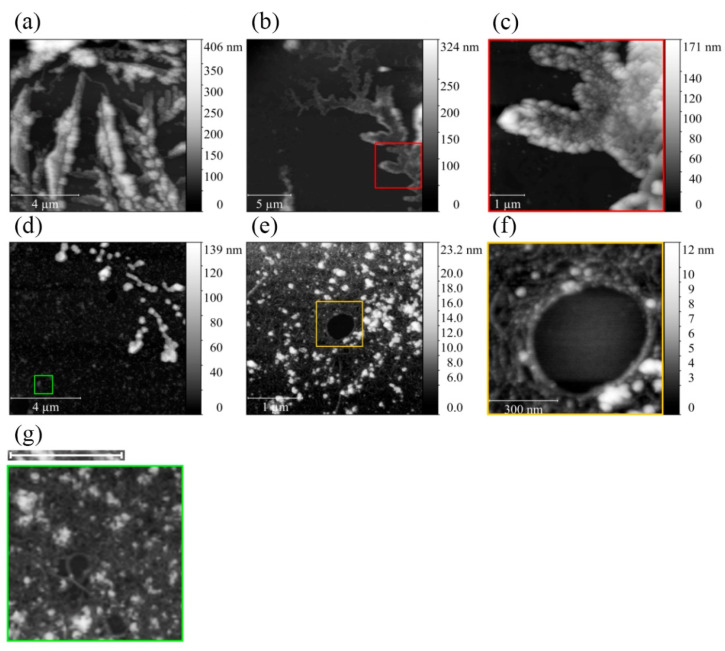
Atomic-force microscopy of SMT_HMW_ aggregates in a solution containing 0.15 M glycine and pH 7.0–7.5. (**a**) SMT_HMW_ after 1 h aggregation. (**b**,**c**) SMT_HMW_ after 24 h aggregation. (**d**,**g**) SMT_HMW_ aggregates after 1 h disaggregation; (**e**,**f**) SMT_HMW_ aggregates after 24 h disaggregation. Scale bar in (**g**), 1 μm. The most representative data, obtained as a result of 3 independent experiments, are given.

**Figure 5 ijms-24-01056-f005:**
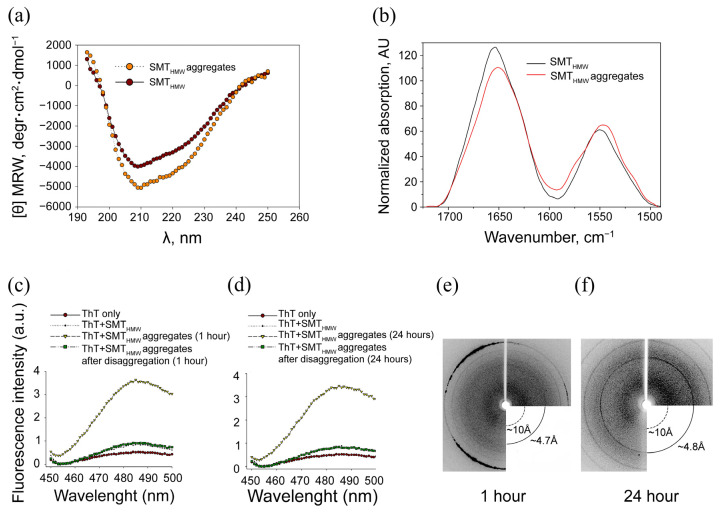
Investigation of the structure of SMT_HMW_ aggregates by various methods. (**a**) CD spectrum of SMT_HMW_; (**b**) FTIR spectra of titin and its aggregates at 20 °C. Protein concentration was 10 mg/mL; (**c**,**d**) thioflavin T (ThT) staining of SMT_HMW_ aggregates formed in a solution containing 0.15 M glycine–KOH at pH 7.0–7.5. (**c**) 1 h formation of aggregates; (**d**) 24 h formation of aggregates; (**e**,**f**) X-ray diffraction of SMT_HMW_ aggregates formed after 1 h (**e**) and 24 h (f).

**Figure 6 ijms-24-01056-f006:**
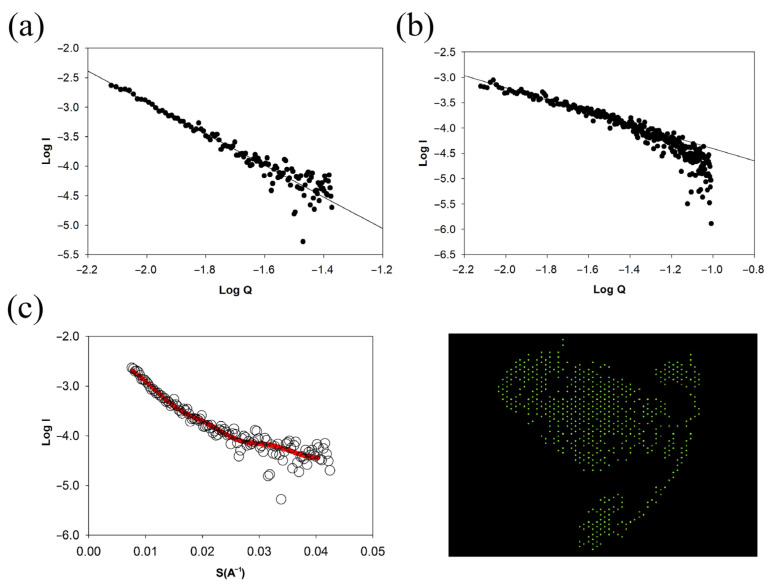
Investigation of the structure of SMT_HMW_ aggregates by small-angle X-ray scattering. (**a**) The scattering curve for molecular SMT_HMW_ in logarithmic representation log *I*–log *Q*, where *I* is the scattering intensity in 0.6 M KCl, *c* = 0.5 mg/mL (non-aggregated protein form), *Q* is the scattering vector, tan *A* = −2.7; (**b**) SMT_HMW_ in 0.15 M Gly (aggregated); *Q*, scattering vector; tan *A* = −2.65; (**c**) SMT_HMW_ in 0.6 M KCl, *c* = 0.5 mg/mL; dark line, approximation by the DAMMIF program.

**Figure 7 ijms-24-01056-f007:**
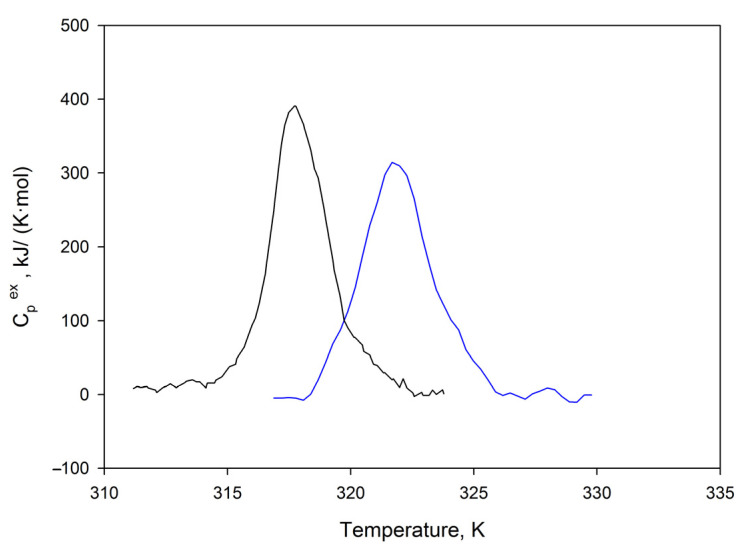
Typical temperature dependence of excess heat capacity of SMT_HMW_ in monomeric (black) and aggregated (blue) forms. Experiments were carried out 2 times for both the monomeric and aggregated SMT_HMW_ forms. The obtained curves coincided in temperature *T*_m_ to an accuracy of 0.1 K. The relative error in determining calorimetric enthalpy, ΔHcal, did not exceed 10%.

**Figure 8 ijms-24-01056-f008:**
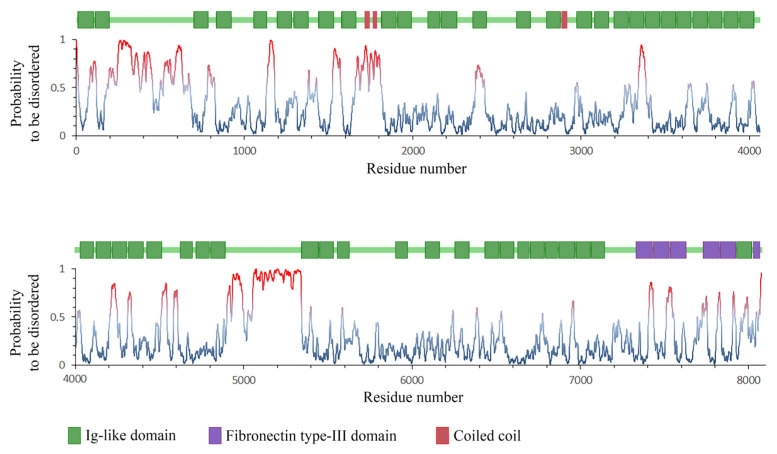
Titin-chicken domain structure schematic aligned with the protein disorder prediction plots by the IsUnstruct program [44]. Probabilities of ≥0.5 mean a disorder.

**Figure 9 ijms-24-01056-f009:**
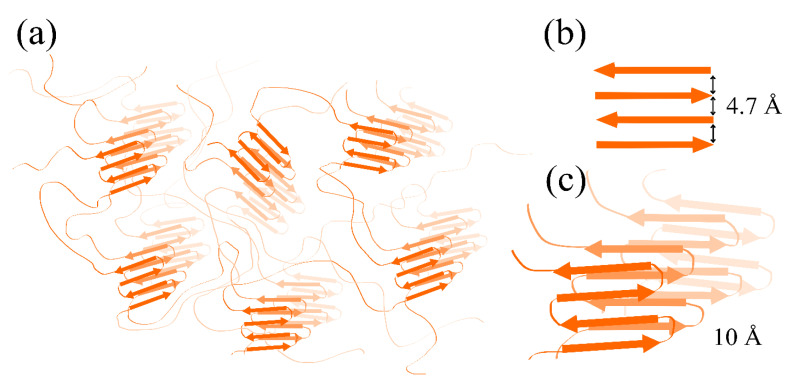
A schematic of the internal structure of aggregates with a cross-β-sheet structure, formed by smooth-muscle titin. (**a**) The proposed structure of SMT aggregates based on circular dichroism, FTIR and X-ray diffraction data, suggesting the presence of a large amount of disorder. (**b**) A beta sheet and the distance of 4.7 Å between the beta strands. (**c**) The beta sheets and the distance of 10 Å between them are shown separately.

**Table 1 ijms-24-01056-t001:** Secondary structure content in chicken titin samples calculated by the FTIR method.

Preparation	α-Helices, %	β-Folds, %	Turns, %	Disordered Structure, %
Molecular SMT_HMW_	26 ± 1.0	25 ± 2.0	11.7 ± 0.6	32 ± 2.0
Aggregated SMT_HMW_	24.6 ± 1.6	26 ± 2.0	10.7 ± 0.4	34 ± 2.0

The results of two experiments are presented. The means and standard deviations are given.

**Table 2 ijms-24-01056-t002:** Identity of the chicken-titin amino acid sequence.

Protein	Domain 1 (for Pair Comparison)	Number of Domains in the Group	Average Length of Domain	Domain 2 (for Pair Comparison)	(Id) Mean Identity, %	Side Standard Dispersion/Deviation of Identity, %	*N*_0_, Number of Domain Pairs with Zero Identity	*N*, Total Number of Domain Pairs	*N*_0_/*N*·100, %
Chicken titin	Ig-like	64	81.4	Ig-like	**20%**	11%	647	4032	16%
	Ig-like	64	81.4	FnIII-like	**3%**	4%	223	384	58%
	Ig-like	64	81.4	none	**2%**	6%	526	704	75%
	FnIII-like	6	84.3	FnIII-like	**33%**	7%	0	30	0%
	FnIII-like	6	84.3	none	**2%**	2%	43	66	65%
	none	11	104.2	none	**3%**	5%	59	110	54%

Identity was calculated by the formula: Id = 2 × *N*_id_/(L_1_ + L_2_). *N*_id_—number of identical residues in the alignment. L—number of residues in the domain. Chicken titin domains were taken from the website: UniProtKB—A6BM71_CHICK. If there was a distance of more than 50 residues between the domains, then a pseudodomain with the signature none FnIII-like = Fibronectin type-III was created. This corresponded to the I-band region of the skeletal muscle sarcomere, which is involved in extension and contraction between the Z-line and the A–I junction [42]. The complete sequence for the smooth muscle isoform of titin or full-sized chicken skeletal titin is not available in the literature.

## Data Availability

All data generated or analyzed in the course of this research (including files of additional information) were incorporated into the article and Appendix A.

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
