# Peer review of "Nonspecific Amyloid Aggregation of Chicken Smooth-Muscle Titin: In Vitro Investigations"

_ijms, 2023, doi:10.3390/ijms24021056_

Round 1

Reviewer 1 Report

The manuscript by Bobylev and co-workers focuses on the characterization of amyloid aggregation of the high-molecular-weight (~1500 kDa) isoform of chicken smooth-muscle titin,. a giant multidomain protein of vertebrate muscles. Using a huge number of various methods (dynamic light scattering, circular dichroism, atomic-force microscopy, transmission electron microscopy, Fourier-transform infrared spectroscopy, differential scanning calorimetry, fluorescence analysis with thioflavin T, X-Ray diffraction, etc.), the authors investigated in detail both the aggregation process leading to formation of titin aggregates and the properties of these aggregates. It was shown that under near-physiological conditions within a relatively short time (~60 min) the protein formed amorphous amyloid aggregates with a quaternary cross-β structure, which partially disaggregated upon increasing the ionic strength above the physiological level (up to 0.6 M KCl. An interesting and uncharacteristic feature of the amyloid aggregation of titin, which distinguishes it from most other amyloid proteins, was that the formation of aggregates was not accompanied by changes in the secondary structure of the protein molecule. It was proposed that this property, i.e. the ability to form amyloid aggregates without changes in the secondary structure of molecules, is a characteristic feature of giant multidomain proteins like titin.  Generally, I agree with this authors’ conclusion.

This is a thorough study with high technical quality. The experiments are carefully performed and clearly described, and the results are well explained. The work is mostly novel, and the manuscript is well written and good-organized. I have no objections to publication of this paper. In my opinion, it could be quite suitable for publication in the International Journal of Molecular Sciences.

Author Response

We are grateful to the Reviewer for a high assessment of our work. Many thanks.

Reviewer 2 Report

In this study, Bobylev et al. use a range of structural biology techniques (e.g. X-ray diffraction, dynamic light scattering, circular dichroism and Fourier-transform infrared spectroscopy to investigate in vitro nonspecific amyloid aggregation of the high-molecular-weight isoform of chicken smooth-muscle titin. They find amorphous amyloid aggregates with a quaternary cross-β structure, formed without changes in the protein secondary structure. The authors conclude that it is not the entire protein but specific domains/segments that are probably involved in the intermolecular interactions during the amyloid aggregation. The authors also suggest that the findings make titin of interest for studying amyloid aggregation in vitro and for expanding our views of the fundamentals of amyloidogenesis. 

The manuscript is overall clear and very well written. An impressive set of advanced methods are used to characterize the proteins and the aggregates formed. The methods are well described and appear to be carefully executed (however see below). The Introduction is generally quite good in giving a fair background to the field. However, an explicit statement of the hypothesis tested and/or the detailed aim(s) of the study is somewhat lacking in the final paragraphs of the Introduction. Other things that I lack in the study and that, in my mind, need to be fully addressed before publication, are the following:

1. Ethical approval of use of animals for protein isolation.

I could not find any information about ethical approval. Please provide.

2. Reproducibility and statistics

This study is comprehensive with regards to the types of experimental analyses of the formed aggregates. However, in order for the study to be reliable and the conclusions fully supported by the data, the authors need to provide evidence that the findings are reproducible. Presently, I perceive (potentially appreciable) weaknesses in this regard. Thus, it is not clear if the experiments are repeated several times on independent samples. If this, for some reasons, would not be needed, strong motivations should be given. Moreover, neither the definition of an independent sample nor the number of such samples used for a given type of experiment, are explicitly mentioned. This should be done for each type of experiment, represented by a figure or a Table. Moreover, throughout the manuscript (in main text, figures and samples), please clarify the meaning of errors/uncertainties (Do they represent standard deviation, SEM, CI or something else?) and (again) the number of independent samples used for their estimation.

3. Quality of the figures.

In several figures, it is difficult to read text and some other information within the figure itself, e.g. due to small font sizes and use of inappropriate font type.

-Fig. 1: Too small font of in-figure-legends but also of axis labelling. Also difficult to see the colors in connection with the text 0 min...60 min in panel A.

Hydrodinamic -> Hydrodynamic on horizontal axis in c.

-Fig. 2: Axes labelling - too small font

-Fig. 5: Font of text inside figures is difficult to read. Please, increase size somewhat and change to sans serif font e.g. Arial.

Minor  (please note that page numbers refer to those given in the footers)

-General: Please explain all abbreviations upon first use despite the list of abbreviations. Please also ensure that the latter list is complete.

-Title: I suggest that the authors (for clarity) change the title “Nonspecific amyloid aggregation of chicken titin: in vitro investigations”  -> “Nonspecific amyloid aggregation of chicken smooth muscle titin: in vitro investigations”

-Abstract: Please spell out “Ig” and “Fn”

-Legend of Fig. 1. “1-h disaggregation”. Did this experiment start with fully aggregated molecules obtained after 1 h? Please specify.

-p11. End of 2nd paragraph. “It should only be noted that this type of aggregation is only a model, since neither functional nor pathological SMTHMW aggregates have been found.” Please clarify! Do you mean that such aggregates have not been found in living cells or…?

Also, I think that there is one "only" to much in this sentence.

-p15, section 5.8, beginning of 2nd paragraph. “Concentration of the protein, 10 mg/mL” -> “Concentration of the protein was 10 mg/mL

-p15. End of section 5.8. “..are adduced” I am not quite sure what you mean?

Author Response

We thank the Reviewer for the great work done, aimed undoubtedly at improving the manuscript. We made some changes at the end of the Introduction, which, in our opinion, better explain the purpose of the study.

Reviewer 2

Other things that I lack in the study and that, in my mind, need to be fully addressed before publication, are the following:

  1. Ethical approval of use of animals for protein isolation.

I could not find any information about ethical approval. Please provide.

Response

Chicken gizzards were acquired from a poultry farm located near the town of Pushchino (Shepilovo Poultry Farm LLC), which has all the necessary licenses for work with animals, in particular, with chickens. Thus, we did not deal directly with animals, but worked only with chicken gizzards. This does not require any ethics commission approval.

Reviewer 2

Reproducibility and statistics

This study is comprehensive with regards to the types of experimental analyses of the formed aggregates. However, in order for the study to be reliable and the conclusions fully supported by the data, the authors need to provide evidence that the findings are reproducible. Presently, I perceive (potentially appreciable) weaknesses in this regard. Thus, it is not clear if the experiments are repeated several times on independent samples. If this, for some reasons, would not be needed, strong motivations should be given. Moreover, neither the definition of an independent sample nor the number of such samples used for a given type of experiment, are explicitly mentioned. This should be done for each type of experiment, represented by a figure or a Table. Moreover, throughout the manuscript (in main text, figures and samples), please clarify the meaning of errors/uncertainties (Do they represent standard deviation, SEM, CI or something else?) and (again) the number of independent samples used for their estimation.

Response

Thank you for this question; indeed, there is some understatement in the manuscript. However, we want to give an assurance that this work has a high reproducibility. The first experiments on the purification of titin from smooth muscles, as well as on the production of aggregates began in 2016. Since then, the protein has been isolated more than 20 times for experiments and has been studied by a large number of methods by co-authors from different laboratories, including international teams. Thus, for several years, results were obtained that were well reproduced. The data obtained by different methods complement one another, which also increases the quality of work. For example, CD data complement FTIR data, despite significant differences in both the principle of operation of these methods and in the process of sample preparation. In the present work, in the methods part, we made a number of explanations regarding the number of repeats.

As for statistics, we can say the following. The features of the object of research and the methods used do not always allow the use of standard statistical methods / approaches to the proper extent, or there is no need for that. For instance, in the case of AFM, statistics is used either in contact mode when measuring specific quantities or to measure the heights of the object of study (http://gwyddion.net/documentation/user-guide-en/statistical-analysis.html\). In our studies, we were only interested in a qualitative assessment of the morphology of the resulting aggregates; and to assess the reproducibility of the data, we analyzed a sample obtained from three isolations, in different areas, and analyzed different areas of the field of study. The same applies to TEM. For reproducibility of the results, we obtained TEM images of titin aggregates from 5 independent protein isolations.

As for DLS, its application is based on measuring the autocorrelation function of light scattering in a protein solution. From these data, we calculate the size distributions of protein formations shown in Fig. 1. The analyzed volume of scattering with a beam cross section of ~100 µm, accounting for the protein concentration used, contains about 10 billion SMT protein molecules, the signal from which is measured. The correlation function signal accumulates over 15 cycles of 15 seconds each  (the way it is described in refs 34, 35). Note that already 5 measurement cycles allow us to obtain adequate size distributions, so that the statistics of signal accumulation from the protein in solution is more than sufficient for each particular experiment. Herewith, we make successive measurements and record smooth temporal changes in size distributions; Figure 1 reflects the distributions corresponding to certain time points. There were 3 independent experiments.

In the case of X-ray diffraction, we carried it out on a single sample obtained from 3 different isolations of the protein. Herewith, it was done with different exposures, at different oscillation angles, and the sample itself was irradiated in different orientations. This is a qualitative method used to confirm the occurrence of reflexes.

To study the binding of titin and its aggregates to the fluorescent dye thioflavin T, we aimed to see the difference between non-aggregated and aggregated forms of the protein. For experiments with disaggregation, 4 independent series of experiments were conducted (4 isolations). This method was used as a control of the amyloid nature of titin aggregates after each isolation (on the whole,  more than 10 isolations).

Below we present several plots for thioflavin T for various independent experiments. The difference in fluorescence intensity is due to the difference in protein concentration, because the protein concentration slightly differed for different isolations (within the range of 0.05–0.2 mg/mL). However, in all cases we were interested in the formation of amyloids, accompanied by an increase in the fluorescence intensity relative to the non-aggregated form of the protein due to the binding of aggregates to the dye.

When studying the secondary structure of titin by the circular dichroism technique, if we have to describe this part in more detail, the CD spectrum in the far UV region was registered in a cuvette with an optical path length of 0.1 cm. Ellipticity was recorded in the wavelength range of 190–250 nm. The spectrum for the reference solution was registered. For a sample under study, the spectrum was taken at an accumulation rate of 10 nm/min. Three repeats of the spectrum were taken for each investigated sample. Data processing and graphical representation were performed in the SigmaPlot program. Based on the absorption spectrum, the exact protein concentration was calculated using the formula: C = ABS280 / l / Ke (where ABS280 is the absorption value at a wavelength of 280 nm, l is the optical path length in cm, Ke is the extinction coefficient).

Three spectra in the far UV region obtained for the investigated sample were averaged and smoothed in the spectropolarimeter software (Spectra Analysis Jasco). A similar procedure was done for the three spectra obtained for the buffer solution. The averaged spectrum of the buffer solution was subtracted from the obtained averaged spectrum of the investigated sample. The value of molar ellipticity [Θ] was calculated by the formula:

[Θ]λ = Θ λ · RMW / l·c,

where Θ λ is the measured value of ellipticity at the wavelength λ, millidegrees; RMW, the average molecular weight of the residue, calculated from the amino acid sequence; l, the optical path length, mm; c, protein concentration, mg/mL.

By the results obtained, a conclusion about the secondary structure of the protein was made. The relative content of the secondary structure and the distribution of peptide bonds’ conformations in proteins were determined by the CD spectrum in the far UV region (range 190–250 nm) using the CONTINLL program of the CDPRO package (https://www.bmb.colostate.edu/cdpro /). The mean root square deviation (RMSD) according to the CDPRO program did not exceed 6%.

Reviewer 2

  1. Quality of the figures.

In several figures, it is difficult to read text and some other information within the figure itself, e.g. due to small font sizes and use of inappropriate font type.

-Fig. 1: Too small font of in-figure-legends but also of axis labelling. Also difficult to see the colors in connection with the text 0 min...60 min in panel A.

Hydrodinamic -> Hydrodynamic on horizontal axis in c.

Response

Changes made, thank you.  

Reviewer 2

-Fig. 2: Axes labelling - too small font

Response

Font size was increased, the figure updated.

Reviewer 2

-Fig. 5: Font of text inside figures is difficult to read. Please, increase size somewhat and change to sans serif font e.g. Arial.

Response

Respective changes made.

Reviewer 2

Minor (please note that page numbers refer to those given in the footers)

Response

Comment taken into account.

Reviewer 2

-General: Please explain all abbreviations upon first use despite the list of abbreviations. Please also ensure that the latter list is complete.

Response

Respective changes to the text of the manuscript and the list of abbreviations were made. In particular, an unnecessary abbreviation was removed: ASI, amino acid sequence identity.

Reviewer 2

-Title: I suggest that the authors (for clarity) change the title “Nonspecific amyloid aggregation of chicken titin: in vitro investigations”  -> “Nonspecific amyloid aggregation of chicken smooth muscle titin: in vitro investigations”

Response

We agree with this comment by the Reviewer. Change made, thank you.

Reviewer 2

-Abstract: Please spell out “Ig” and “Fn”

Response

Done

Reviewer 2

-Legend of Fig. 1. “1-h disaggregation”. Did this experiment start with fully aggregated molecules obtained after 1 h? Please specify.

Response

After 1 h of aggregation in a solution with a low ionic strength, the aggregated protein preparation was again placed under high ionic strength conditions for 1 h (a column buffer; ionic strength, greater than 0.6); the column buffer initially contains the molecular (non-aggregated) form of the protein. The disaggregation conditions are briefly described in Section 5.3 (Conditions for the formation of SMT aggregates): “In disaggregation experiments, SMT aggregates were dialyzed during 1 and 24 h against a column buffer.”. Thus, with respect to the disaggregation: The protein aggregated for 1 h was disaggregated also for 1 h. The protein aggregated for 24 h  was subject to disaggregation for, respectively, 24 h.

Reviewer 2

-p11. End of 2nd paragraph. “It should only be noted that this type of aggregation is only a model, since neither functional nor pathological SMTHMW aggregates have been found.” Please clarify! Do you mean that such aggregates have not been found in living cells or…?

Also, I think that there is one "only" to much in this sentence.

Response

No, in living cells titin aggregates have not been found; the second “only was removed; “in living cells” was added at the end of the sentence:

“… neither functional nor pathological SMTHMW aggregates have been found in living cells.”  

Reviewer 2

-p15, section 5.8, beginning of 2nd paragraph. “Concentration of the protein, 10 mg/mL” -> “Concentration of the protein was 10 mg/mL“

Response

Correction made.

Reviewer 2

-p15. End of section 5.8. “…are adduced” I am not quite sure what you mean?

Response

Changed to “… are given.” The sentence now reads as follows:

“The standard deviations of the values of the secondary structure elements in the protein are given.” Thanks.

Round 2

Reviewer 2 Report

I am generally happy with the changes made by the authors. Based on descriptions in their response, I also feel convinced that the findings are reproducible. However, this should also be made clear in the paper, not only in the Response to Reviewers. Thus, I suggest that the authors transfer most of their response to my comments on Statistics and Reproducibility to the paper. First, specific information of the number of repetitions (protein isolations, replicate experiments etc) should be given under respective sections in the Materials and Methods. Additionally, some discussion of other aspects of the reproducibility would be of value in the Discussions. 

Author Response

Dear colleagues,

Hopefully, we have fulfilled the Reviewer’s wishes and recommendations to a maximum. This updated version accepts earlier changes and makes new changes as recommended in the last round.

The new changes / additions are in the following lines:

165-166, 195-196, 207-208, 303-306, 374-375, 497-498, 539-540, 549-550, 573-574, 581-599, 620-621, 629-631, 635-636, 675.

Once again, many thanks for the thorough and thoughtful reviewing.

A Merry Christmas week and a Happy New Year!

Sincerely,

Alexander Bobylev and Ivan Vikhlyantsev on behalf of the co-authors